# Universal Pyramid Adversarial Training for Improved ViT Performance

## Abstract

Recently, Pyramid Adversarial training (Herrmann et al., 2022) has been shown to be very effective for improving clean accuracy and distribution-shift robustness of vision transformers. However, due to the iterative nature of adversarial training, the technique is up to 7 times more expensive than standard training. To make the method more efficient, we propose Universal Pyramid Adversarial training, where we learn a single pyramid adversarial pattern shared across the whole dataset instead of the sample-wise patterns. With our proposed technique, we decrease the computational cost of Pyramid Adversarial training by up to 70% while retaining the majority of its benefit on clean performance and distribution-shift robustness. In addition, to the best of our knowledge, we are also the first to find that universal adversarial training can be leveraged to improve clean model performance.

## 1 Introduction

Human intelligence is exceptional at generalizing to previously unforeseen circumstances. While deep learning models have made great strides with respect to clean accuracy on a test set drawn from the same distribution as the training data, a model's performance often significantly degrades when confronted with distribution shifts that are qualitatively insignificant to a human. Most notably, deep learning models are still susceptible to adversarial examples (perturbations that are deliberately crafted to harm accuracy) and out-of-distribution samples (images that are corrupted or shifted to a different domain).

Adversarial training has recently been shown to be a promising avenue for improving both clean accuracy and robustness to distribution shifts. While adversarial training was historically used for enhancing adversarial robustness, recent works (Xie et al., 2020; Herrmann et al., 2022) found that properly adapted adversarial training regimens could be used to achieve state-of-the-art results (at the time of publication) on Imagenet (Xie et al., 2020) and out-of-distribution robustness (Herrmann et al., 2022).

However, both proposed techniques (Herrmann et al., 2022; Xie et al., 2020) use up to 7 times the standard training compute due to the sample-wise and multi-step procedure for generating adversarial samples. The expensive cost has prevented it from being incorporated into standard training pipelines and more widespread adoptions. In this paper, we seek to improve the efficiency of the adversarial training technique so that it can become more accessible for practitioners and researchers.

Several prior works (Shafahi et al., 2019; Zhang et al., 2019; Mei et al., 2022; Zheng et al., 2020; Wong et al., 2020) have proposed methods to increase the efficiency of adversarial training in the context of adversarial robustness, where they try to make models robust to deliberate malicious attacks. Shafahi et al. (2019) proposed reusing the parameter gradient for training during the sample-wise adversarial step for faster convergence. Later, Wong et al. (2020) proposed making adversarial training more efficient with a single-step adversary rather than the expensive multi-step adversary. However, all prior works focus on the efficiency trade-off concerning adversarial robustness rather than clean accuracy or out-of-distribution robustness. In the setting of adversarial robustness, one often assumes a deliberate and all-knowing adversary. Security is crucial, yet in reality, deep learning systems already exhibit a significant number of errors without adversaries, such as self-driving cars making mistakes in challenging environments. Consequently, clean accuracy and

robustness to out-of-distribution data are typically prioritized in most industrial settings. Yet, few works seek to improve the efficiency trade-off for the out-of-distribution metric. Our work aims to fill this gap.

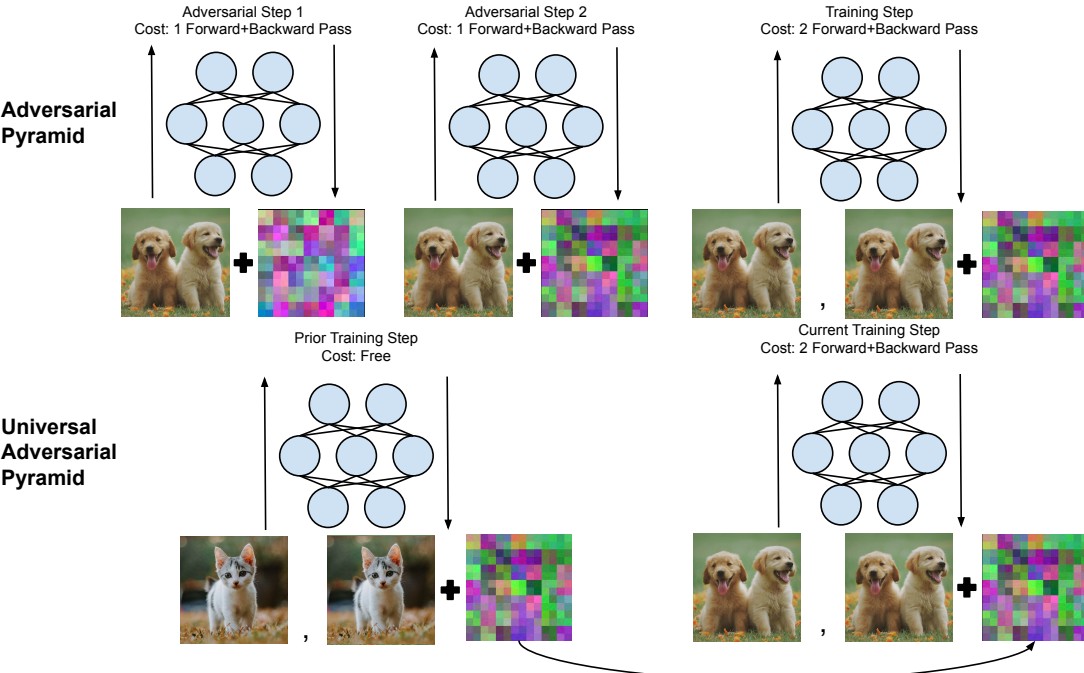

Figure 1: A graphical illustration of how our Universal Pyramid Adversarial training is more efficient compared to Pyramid Adversarial training. In the specific examples, the two-step Pyramid Adversarial training requires 4 forward and backward passes for a single example, where 2 passes are used for generating the adversarial sample and 2 additional passes are used for training. On the other hand, our proposed approach only requires 2 forward and backward passes. These two passes are needed because the batch size has been doubled since we optimize both the clean and the adversarial objectives.

By shifting the context from adversarial robustness to clean accuracy and out-of-distribution robustness, we can free ourselves from certain constraints, such as the need to train the model on *sample-wise* adversaries, which is very expensive to compute. Instead, we can leverage *universal* perturbations, which are shared across the whole dataset. By leveraging this simple idea, we can generate adversarial samples for *free* while getting more performance improvement on clean accuracy compared to prior work (Herrmann et al., 2022).

In this paper, we focus our experiments on the Vision Transformer architecture (Dosovitskiy et al., 2020). We focus on this architecture as it is the most general and scalable architecture that applies to many domains, including vision, language, and audio, while simultaneously achieving SOTA on many of them. We believe focusing on this architecture will lead to more valuable techniques for the community.

In summary, here are our three main contributions:

- We propose Universal Pyramid Adversarial training that is 70% more efficient than the multi-step approach while increasing ViT's clean accuracy more than Pyramid Adversarial training.

- We evaluate our technique on 5 out-of-distribution datasets and find that Universal Pyramid Adversarial training effectively increases the distributional robustness and is competitive with Pyramid Adversarial Training while being efficient.

- To the best of our knowledge, we are the first to identify universal adversarial training as a viable technique for improving clean performance and out-of-distribution robustness on Imagenet 1K. In our ablations, , we found that the pyramid structure is critical for the performance gain and plain

universal adversarial training is detrimental to performance, unlike Herrmann et al. (2022), which found both instance wise adversarial training and pyramid adversarial training to be beneficial.

## 2 Related Work

Improving the efficiency of adversarial training has been widely studied (Shafahi et al., 2019; Zhang et al., 2019; Zheng et al., 2020; Wong et al., 2020), but they have mainly been in the context of adversarial robustness. Shafahi et al. (2019) proposed reusing the parameter gradients from the adversarial step. By reusing the free parameter gradients for training, they were able to achieve much faster convergence. Even though the proposed approach was much more efficient, Shafahi et al. (2019) could not reach the same robustness level as the original multi-step training on Imagenet-1K. Zhang et al. (2019) proposed making the iterative attack cheaper by updating the noise based on the Hamiltonian functions of the first few layers. Wong et al. (2020) proposed using a single-step adversary for training instead of a multi-step adversary. They found that random initialization and early stopping could prevent adversarial over-fitting, where label leakage happens from using adversaries with fewer steps. Zheng et al. (2020) proposed reusing the adversarial perturbations between epochs with the observation that adversarial noises are often transferable. The downside of the method is that the memory requirement grows with the data size, which can be quite large given the size of modern datasets. We differ from all the prior works in that we aim to investigate the efficiency gain of adversarial training in the context of clean accuracy. By focusing on clean accuracy and out-of-distribution robustness, we gained more flexibility concerning the formulation of the min-max problem.

Xie et al. (2020) was the first paper that showed adversarial training could improve clean performance of convolutional networks. To achieve this, Xie et al. (2020) employed split batchnorms (AdvProp) for adversarial and clean samples. They argued that clean and adversarial samples have very different distributions and that split batchnorms are needed to make optimization easier. Before Xie et al. (2020), the community commonly believed that adversarial training leads to a decrease in clean accuracy (Madry et al., 2018).

In a similar line of work, Mei et al. (2022) proposed a faster variant of AdvProp (Xie et al., 2020) that makes the training speed comparable to standard training while retaining some of Advprop's benefits. Even though the proposed method is more efficient, it substantially trades off the performance of the original multi-step method. Our work is similar to Mei et al. (2022) because we also focus on improving the efficiency of the adversarial training process, but we differ in that we focus on ViT architecture with Pyramid Adversarial training where their proposed approach is not applicable. Also, we can achieve a performance gain that is *comparable or better* than the multi-step approach, whereas the previous method trades off performance for efficiency.

Several recent approaches have shown that adversarial training can be used to improve the performance of vision transformers. Bai et al. (2022) showed that ViT relies more on low-frequency signals than high-frequency signals. By adversarially training the model on high-frequency signals, Bai et al. (2022) further boosted ViT's performance. Mao et al. (2022) showed that by converting images to discrete tokens, adversarial training could further increase the performance of ViTs. Later, Herrmann et al. (2022) showed that by incorporating the pyramid structure into standard adversarial training, they could boost the performance of ViT, where the split batchnorms idea introduced in Xie et al. (2020) were not directly applicable to ViT models. In our work, we focus on the Pyramid Adversarial training technique proposed by Herrmann et al. (2022) since it is the best-performing method that achieves SOTA on multiple fronts while being applicable to ViT, a more modern architecture. The main drawback of Herrmann et al. (2022) is the significantly higher training time which can go up to 7x of the standard training time. In this work, we propose Universal Pyramid Adversarial training to improve the efficiency of Pyramid Adversarial training while retaining its effectiveness.

While universal adversarial samples have been used in prior work Shafahi et al. (2020) for training to defend against universal adversarial attacks, our proposed approach differs from them in that it leverages these samples to improve clean model performance. Benz et al. (2021) finds that universal perturbations tend to slide images into some classes more than others. They find that by updating universal perturbations in a class-wise manner, they can achieve better robustness compared to Shafahi et al. (2019). Both prior works Shafahi et al. (2020); Benz et al. (2021) show that universal adversarial training consistently decreases the

performance of the model, similar to standard adversarial training. Our ablation study shows that incorporating both clean loss and pyramid structures are crucial for the performance gain observed in Universal Pyramid Adversarial training. Without our proposed modifications, universal adversarial training consistently decreases the clean performance of the model. To the best of our knowledge, we are the first to show that universal adversarial training can be leveraged for improved model performance.

## 3    Method

In this section, we will go over the formulation of the proposed adversarial training objective, the pyramid structure that we leveraged from Herrmann et al. (2022), and our more efficient Universal Pyramid Adversarial training.

### 3.1    Adversarial Training

Adversarial training remains one of the most effective methods for defending against adversarial attacks Bai et al. (2021). Adversarial training is aimed at solving the following min-max optimization problem:

$$\min_{\theta} E_{(x,y) \sim D} \left[ \max_{\delta \in B} L(f(x + \delta; \theta), y) \right],\tag{1}$$

where $\theta$ is the model parameter, $\delta$ is the adversarial perturbation, $L$ is the loss function, $D$ is the data distribution, and $B$ is the constraint for the adversarial perturbation, which is often an $\ell_{\infty}$ ball. The inner objective seeks to find an adversarial perturbation within the constraint, and the outer objective aims to minimize the worst-case loss by optimizing the model parameters. While the method effectively improves robustness to adversarial attacks, it often reduces clean performance. However, the loss of clean performance is often not acceptable for most practical applications.

Since our goal is to improve performance as opposed to worst case robustness, we train the model on the following formulation (similar to Xie et al. (2020); Herrmann et al. (2022)) instead where the clean loss is optimized in addition to the adversarial loss:

$$\min_{\theta} E_{(x,y) \sim D} \left[ L(f(x; \theta), y) + \lambda \max_{\delta \in B} L(f(x + \delta; \theta), y) \right].\tag{2}$$

Here, the $\lambda$ controls the trade-off between adversarial and clean loss. However, adding clean loss alone is often not sufficient for improving the performance of the model Xie et al. (2020). Additional techniques such as split batchnorms Xie et al. (2020) and pyramid structures Herrmann et al. (2022) are necessary for performance gain.

The main problem with the adversarial training formulation is that the inner maximization is often expensive to compute, requiring several steps to approximate accurately Madry et al. (2018). Specifically, for each iteration of the adversarial step, one needs a full forward and backward pass on all of the examples in a batch (see Figure 1). For example, if five steps are used, which is the setting in both Xie et al. (2020); Herrmann et al. (2022), then five forward and backward passes are needed. The generation of adversarial samples is already five times more expensive than regular training. In addition, one needs to use both the clean and the generated adversarial samples for training, so now we have doubled the batch size. The larger batch size increases the cost by another factor of two. When training with a 5-step adversary, the total computational cost will be seven times more expensive than standard training. In Section 3.3, we describe our proposed Universal Pyramid Adversarial training, where we can substantially reduce the computational cost.

### 3.2    Pyramid Structure

Adversarial training alone even when coupled with the clean loss does not typically increase performance of the model Xie et al. (2020). In order to increase clean accuracy, certain techniques have to be used. Here we leverage the pyramid structure from Herrmann et al. (2022). Herrmann et al. (2022) aimed to endow the

adversarial perturbation with more structure so that the adversary can make larger edits without changing an image's class. The pyramid adversarial noise is parameterized with different levels of scales as follows:

$$\delta = \sum_{s \in S} m_s \cdot C_B(\delta_s), \tag{3}$$

where $C_B$ clips the noise within the constraint set $B$, $S$ is all of the scales used, $m_s$ is the multiplicative constant, and $\delta_s$ is perturbation at scale $s$. For the $\delta_s$ at a given scale, $s \times s$ number of pixels within a square tile share a single parameter giving greater structure to the noise. Since the larger scale can often tolerate more changes, larger $m_s$ at the coarser scales allow us to update the coarser noises more quickly relative to the granular noise.

### 3.3 Universal Pyramid Adversarial Training

While Pyramid Adversarial training Herrmann et al. (2022) is effective at increasing clean model performance, it is seven times more expensive compared to standard training. To address this, we propose Universal Pyramid Adversarial training, an efficient adversarial training approach to improve model performance on clean and out-of-distribution data. Our proposed approach learns a universal adversarial perturbation with pyramid structure, thus unifying both the effectiveness of Pyramid Adversarial training and the efficiency of universal adversarial training Shafahi et al. (2020). Specifically, we attempt to solve the following objective:

$$\min_{\theta} \max_{\delta \in B} \mathop{E}_{(x,y) \sim D} \left[ L(f(x; \theta), y) + \lambda L(f(x + \delta; \theta), y) \right]. \tag{4}$$

With this objective, we only have to solve for a *single* universal adversarial pattern that can be shared across the whole dataset, and we do not have to optimize a new adversary for each sample. Even though the objective looks similar, they are not the same. Due to Jensen's inequality, Equation 4 is always strictly upper-bounded by Equation 2. We have described the complete method in Algorithm 1. This yields up to 70% saving compared to the 5-step sample-wise approach (see Table 1). Further, we update the universal adversarial pattern during the backward pass of training, where we can get the gradients of $\delta$ for free (see Figure 1 for an illustration of how universal adversarial training can help save compute). However, since we still need to train the model on twice the number of samples, our proposal is still twice as expensive as standard training, but it is already 33% cheaper than the fastest (one-step) sample-wise adversarial training approach.

More concretely, the generation step for the one-step sample-wise adversarial training costs a single forward-backward pass, and the training step is twice as expensive as standard training. Overall, one-step sample-wise adversarial training is 3x the cost of standard training making our method 33% faster. This is because in case of the one-step adversarial training, the gradient from the first generation step cannot be reused because the patterns are randomly initialized and the induced gradient is different from the clean training gradient.

### 3.4 Radius Schedule

In our experiments, we find that a radius schedule occasionally benefits performance. The radius dictates the extent to which an adversary is permitted to alter the image, as measured by the $\ell_\infty$ distance between the original and perturbed images. A larger radius permits greater perturbations, thus strengthening the adversary, while a smaller radius restricts the perturbation, rendering the adversary weaker. We propose this schedule as we observe that a more aggressive/larger radius tends to promote faster convergence at the beginning, but these images are very far out of distribution, resulting in poor performance. By using a linearly decreasing radius schedule, we are sometimes able to get a considerable performance boost while maintaining fast convergence. Precisely, we calculate the radius at a given epoch as follows

$$r(e) = r_{\text{start}} + (r_{\text{end}} - r_{start}) \frac{\max(e - e_{start}, 0)}{e_{end} - e_{start}}, \tag{5}$$

where $r_{start}, r_{end}$ are the starting and ending radius with $r_{start} > r_{end}$, $e_{start}, e_{end}$ are the starting and ending epochs for the radius schedule, and $r(e)$ is the radius at a given epoch $e$.

---

**Algorithm 1** Universal Pyramid Adversarial Training

---

$\tau$ is the step size of adversarial attack
$\lambda$ controls the regularization strength
Initialize $\delta_s$ as zeros
**for** epoch $= 1 \ldots N_{ep}$ **do**
    **for** $x, y \in D$ **do**
        $\delta \leftarrow \sum_{s \in S} m_s C_B(\delta_s)$
        $\mathcal{L}(x, y, \delta; \theta) \leftarrow \ell(x, y; \theta) + \lambda \ell(x + \delta, y; \theta)$
        $\theta \leftarrow \theta - lr \cdot \nabla_\theta \mathcal{L}(x, y, \delta; \theta)$
        \\Update model parameters with some optimizer
        **for** $s \in S$ **do**
            $\delta_s \leftarrow \delta_s + \tau \cdot sign(\nabla_{\delta_s} \mathcal{L}(x, y, \delta; \theta))$
            \\Update noise with the free gradients
        **end for**
    **end for**
**end for**

---

| Augmentation | Method | Radius | Radius Schedule | # of Steps | Training Time (hrs) | Top 1 Accuracy | Gain |
|---|---|---|---|---|---|---|---|
| | Baseline | - | - | - | 12.7 | 72.90% | - |
| | Pyramid Adversarial TrainingHerrmann et al. (2022) | 6/255 | - | 1 | 36.5 | 73.52% | 0.62% |
| | | 6/255 | - | 2 | 49.3 | 73.95% | 1.05% |
| Weak | | 6/255 | - | 3 | 61.8 | 74.68% | 1.78% |
| | | 6/255 | - | 4 | 75.4 | 74.80% | 1.90% |
| | | 6/255 | - | 5 | 88.9 | 74.18% | 1.28% |
| | Universal Pyramid Adversarial Training | 8/255 | Yes | - | 26.6 | 74.87% | **1.97%** |
| | | 8/255 | - | - | 26.6 | 74.57% | 1.67% |
| | Baseline | - | - | - | 12.7 | 79.85% | - |
| | Pyramid Adversarial TrainingHerrmann et al. (2022) | 6/255 | - | 1 | 36.5 | 79.46% | -0.38% |
| | | 6/255 | - | 2 | 49.3 | 79.87% | 0.03% |
| Strong | | 6/255 | - | 3 | 61.8 | 80.19% | 0.35% |
| | | 6/255 | - | 4 | 75.4 | 80.04% | 0.22% |
| | | 6/255 | - | 5 | 88.9 | 80.10% | 0.26% |
| | Universal Pyramid Adversarial Training | 8/255 | Yes | - | 26.6 | 80.15% | 0.31% |
| | | 8/255 | - | - | 26.6 | 80.28% | **0.44%** |

Table 1: Comparison of the effectiveness of universal and sample-wise pyramid adversarial training. Universal Pyramid Adversarial training improves performance compared to sample-wise Pyramid Adversarial training while being much more efficient.

## 4 Experiments

### 4.1 Experimental Set-up

In all of our experiments, we focus on the training setup in Beyer et al. (2022) since it allows us to achieve a competitive 79.8% on Imagenet-1K with a ViT-S/16. The setup allows us to study ViT in a computationally feasible setting.

Following Beyer et al. (2022), we use the AdamW optimizer, with a batch size of 1024, a learning rate of 0.001 with a linear warm-up for the first 8 epochs, and weight decay of 0.1. We train the model for a total of 300 epochs across all settings. For augmentation, we apply a simple inception crop and horizontal flip. For experiments with strong data augmentation, we apply RandomAugment Cubuk et al. (2020) of level 10 and MixUp Zhang et al. (2017) of probability 0.2. We used strong data augmentation in all of our experiments except for the first part of experiments in Table 1.

For Pyramid Adversarial training, following Herrmann et al. (2022) we use $S = [32, 16, 1]$, $M = [20, 10, 1]$, and radius of 6/255. For step size, we simply divide the radius by the number of steps used. For our proposed Universal Pyramid Adversarial training, we use the same $M$ and $S$ as Pyramid Adversarial training, but

| Method | Radius | Training Time (hrs) | Clean ↑ | C ↓ | A ↑ | Rendition ↑ | Sketch ↑ | Stylized ↑ |
|---|---|---|---|---|---|---|---|---|
| Universal Pyramid Adversarial Training | 8/255 | 26.6 | 80.28% | 41.02% | 29.08% | 35.45% | 16.89% | 17.74% |
| Gain/Loss Relative to | | | | | | | | |
| - Regular Training | - | 12.7 | 0.44% | -1.16% | 1.56% | 1.80% | 1.81% | 0.68% |
| - Pyramid Adversarial Training 1 steps | 6/255 | 36.5 | 0.82% | 0.62% | 2.60% | 0.11% | 0.33% | -1.26% |
| - Pyramid Adversarial Training 2 steps | 6/255 | 49.3 | 0.41% | 1.25% | 0.59% | -0.71% | -0.06% | -2.21% |
| - Pyramid Adversarial Training 3 steps | 6/255 | 61.8 | 0.09% | 2.10% | -1.01% | -2.29% | -0.77% | -3.22% |
| - Pyramid Adversarial Training 4 steps | 6/255 | 75.4 | 0.24% | 1.80% | -1.25% | -2.95% | -1.13% | -3.28% |
| - Pyramid Adversarial Training 5 steps | 6/255 | 88.9 | 0.18% | 2.17% | -0.96% | -3.17% | -0.91% | -3.47% |
| Universal Pyramid Adversarial training | 12/255 | 26.6 | 80.04% | 40.37% | 28.21% | 37.05% | 17.42% | 19.53% |
| Gain/Loss Relative to | | | | | | | | |
| - Regular Training | - | 12.7 | 0.20% | -1.81% | 0.69% | 3.39% | 2.34% | 2.48% |
| - Pyramid Adversarial Training 1 steps | 6/255 | 36.5 | 0.58% | -0.02% | 1.73% | 1.70% | 0.87% | 0.53% |
| - Pyramid Adversarial Training 2 steps | 6/255 | 49.3 | 0.17% | 0.60% | -0.28% | 0.88% | 0.48% | -0.41% |
| - Pyramid Adversarial Training 3 steps | 6/255 | 61.8 | -0.15% | 1.46% | -1.88% | -0.70% | -0.23% | -1.42% |
| - Pyramid Adversarial Training 4 steps | 6/255 | 75.4 | 0.00% | 1.15% | -2.12% | -1.36% | -0.59% | -1.48% |
| - Pyramid Adversarial Training 5 steps | 6/255 | 88.9 | -0.06% | 1.52% | -1.83% | -1.58% | -0.38% | -1.67% |

Table 2: Evaluating models trained with Universal Pyramid Adversarial training (across two radius) on 5 additional out-of-distribution datasets and comparing them with regular and Pyramid Adversarial training. Universal Pyramid Adversarial training consistently boosts out-of-distribution robustness and is competitive with 1-step and 2-step Pyramid Adversarial training. We report the gain/loss relative to our method, i.e., the values in the colored cells are calculated by the following formula: (our method - the alternative method). For all columns except the Imagenet-C column, positive numbers mean that our method performs better and vice versa. We report mean Corruption Error (mCE) for Imagenet-C where lower is better. For the rest of the datasets, we simply report the top-1 accuracy. The complete table can be found in the Appendix.

with a radius of 8/255. When a radius schedule is used, we linearly decrease the radius by 90% in increments starting from epoch 30.

In addition to Imagenet-1K, we also evaluated our models on five out-of-distribution datasets: Imagenet-C (Hendrycks & Dietterich, 2019), Imagenet-A (Hendrycks et al., 2021), Imagenet-Rendition, Imagenet Sketch (Wang et al., 2019), and Stylized Imagenet (Geirhos et al., 2019). Using a diverse set of out-of-distribution datasets, we can more thoroughly evaluate the model's robustness to unexpected distribution shifts.

## 4.2 Experimental Results

Overall, we find that Universal Pyramid Adversarial training effectively increases the clean accuracy and out-of-distributional robustness similar to the original Pyramid Adversarial training while being much more efficient.

**Clean Accuracy**  Here we first analyze the effectiveness of our proposed Universal Pyramid Adversarial training when applied to ViT with weak data augmentation. Pyramid Adversarial training, as expected, significantly increases the performance of the ViT by up to 1.9% (Table 1) when a 4-step attack is used. As we increase the step count to 5, the benefit of Pyramid Adversarial training starts diminishing. On the other hand, our Universal Pyramid Adversarial training increased the performance even further. With the radius schedule, we can obtain a performance increase of 1.97%, exceeding the performance benefit of the Pyramid Adversarial training for all step counts. Without the radius schedule, we still obtain a competitive gain of 1.67%. In addition to better performance, our method is much more efficient than the original Pyramid Adversarial training. In Table 1, we also reported the training time of each method on 8 Nvidia A100 GPUs in hours, and our approach is 70% faster compared to the 5-step Pyramid Adversarial training.

To further verify our proposal's effectiveness, we analyze our method's performance when coupled with strong data augmentations. The setting with strong data augmentation is a more challenging setting since all components of the training pipelines are heavily tuned. It is worth noting that our baseline ViT-S performs comparably with the baseline ViT-B/16 in Herrmann et al. (2022); Steiner et al. (2021). Again, we continue to see the benefit of Universal Pyramid Adversarial training compared to standard training.

| | Radius 2/255 | 4/255 | 6/255 | 8/255 | 10/256 | 12/256 |
|---|---|---|---|---|---|---|
| Baseline | 79.85% | | | | | |
| Universal Adversarial Training (w/o Pyramid Structure) | 79.73% | 79.81% | 79.64% | 79.59% | 79.65% | 79.75% |
| - Gain Relative to Baseline | -0.12% | -0.04% | -0.21% | -0.26% | -0.20% | -0.10% |
| Universal Pyramid Adversarial Training | 79.89% | 80.23% | 80.15% | 80.28% | 80.13% | 80.04% |
| - Gain Relative to Baseline | 0.04% | 0.38% | 0.30% | 0.43% | 0.28% | 0.19% |

Table 3: Comparing universal adversarial training with and without the pyramid structure. We find that the pyramid structure is indeed crucial for the performance gain observed. Without the pyramid structure, universal adversarial training consistently decreases the model's performance relative to the baseline.

| Method | Radius | Top 1 Accuracy | Gain |
|---|---|---|---|
| Baseline | - | 79.84% | - |
| Uni. Pyramid Adv. | 2/255 | 79.89% | 0.05% |
| | 4/255 | 80.23% | 0.39% |
| | 6/255 | 80.15% | 0.31% |
| | 8/255 | 80.28% | **0.44%** |
| | 10/255 | 80.13% | 0.29% |
| | 12/255 | 80.04% | 0.20% |

Table 4: Universal Pyramid Adversarial training consistently increases the performance of ViT across a range of hyperparameters but achieves the best performance at radius 8/255.

In the more challenging setting, we see less benefit from both approaches, as expected. As we increase the number of steps for the Pyramid Adversarial training, the accuracy first increases, reaching the maximum at 3 steps, and starts decreasing with 4 or more steps. On the other hand, our Universal Pyramid Adversarial training achieves more performance gain than all of the step count tested while being more efficient.

**Out-of-Distribution Robustness** We found that Universal Pyramid Adversarial training effectively increases models' out-of-distribution robustness and is comparable to 1-step and 2-step Pyramid Adversarial training. In Table 2, we see that our Universal Pyramid Adversarial training consistently increases models' performance on all five out-of-distribution datasets relative to the baseline. When compared with Pyramid Adversarial training, we find that Universal Pyramid Adversarial training with a radius of 12/255 consistently improves performance with respect 1-step Pyramid Adversarial training and is comparable with the 2-step Pyramid Adversarial training. Note that both 1-step and 2-step Pyramid Adversarial training are already 50% and 100% more expensive than our proposed Universal Pyramid Adversarial training. However, unlike the case of clean accuracy, Universal Pyramid Adversarial training still underperforms relative to the more costly Pyramid Adversarial training with 3 or more steps.

### 4.3 Ablations

In this section, we ablated several components of Universal Pyramid Adversarial training, including its sensitivity to the selected radius, the importance of the pyramid structure, and the benefit of incorporating clean loss.

**Sensitivity to Radius** Hyperparameter sensitivity is crucial for a method's practicality, and we find that our Universal Pyramid Adversarial training is consistent and stable with respect to the selected radius. In Table 4, we see that Universal Pyramid Adversarial training consistently increases the model's performance across a wide range of radii from 2/255 to 12/255. This consistency is important because it allows us to benefit from the method without finely tuning the radius. We also find that the performance varies in a predictable upside-down U-shape. As we increase the radius, we see that the performance steadily increases

|  | Top 1 Accuracy |
| --- | --- |
| Baseline | 79.84% |
| Uni. Pyramid Adv. | |
| w/o clean loss | 78.08% (-1.76%) |
| w/ clean loss | 80.28% (+0.44%) |

Table 5: Comparing Universal Pyramid Adversarial training with and without adding the clean loss. We found that the addition of clean loss is critical for performance improvements.

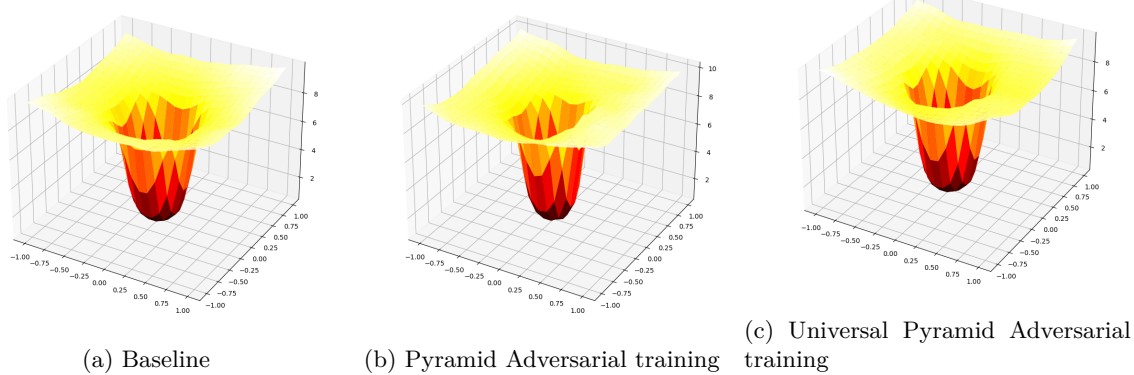

(a) Baseline    (b) Pyramid Adversarial training    (c) Universal Pyramid Adversarial training

Figure 2: Analyzing the loss landscape of the final trained models. We employed the filter normalization method from Li et al. (2017) for visualization. Pyramid Adversarial training seems to induce minima that are sharper compared to the baseline and Universal Pyramid Adversarial training. However, despite landing the model in sharper minima, Pyramid Adversarial training still produces a better performing model than the baseline. On the other hand, Universal Pyramid Adversarial training does not seem to change the sharpness of minima while attaining the best performance. This observation shows that both adversarial pyramid approaches do not improve performance through flattening the loss landscape.

until radius 8/255 after which the performance decreases. The way that performance changes with respect to the radius gives practitioners a clear signal on whether to increase or decrease the radius and makes hyperparameter tuning easier.

**Pyramid Structure**  We also found that the pyramid structure is crucial for performance gain with our proposed universal adversarial approach. We experimented with naively combining clean loss with universal adversarial training as in Shafahi et al. (2020) as an additional regularizer. However, in Table 3, we see that without using the pyramid structure, the model consistently performs worse after adding the adversarial samples.

**Clean Loss**  In addition to using pyramid structure, we found that incorporating clean loss to the Universal Pyramid Adversarial training is vital to obtain the performance gain we see. In Table 5, we see that removing the clean loss results in a performance decrease of 1.76%, making the model much worse than the baseline.

### 4.4 Analysis

In this section, we try to understand the similarities between universal and sample-wise adversarial pyramid training, given their similar benefits and formulation. We analyzed the attack strength, the noise pattern, and the loss landscape and found that despite their similarities, they are quite different in many aspects. The findings point to the need to further understand the mechanism that both universal and sample-wise adversarial pyramid training use to increase model performance.

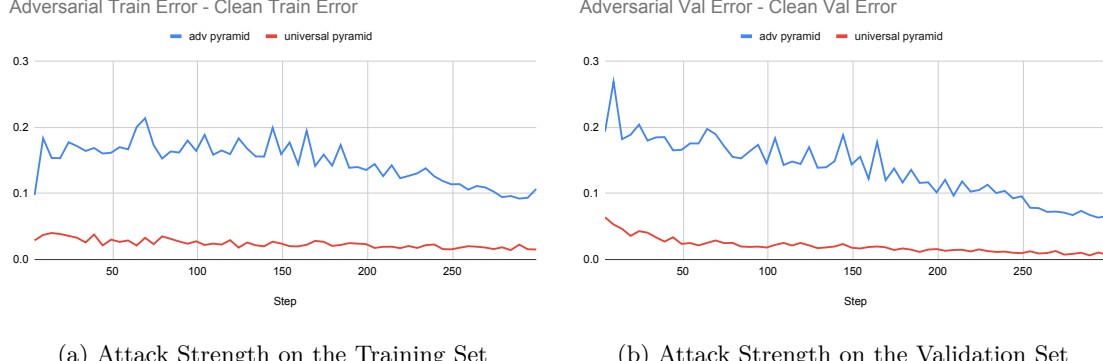

(a) Attack Strength on the Training Set        (b) Attack Strength on the Validation Set

Figure 3: Comparing the strength of the universal adversary with the sample-wise adversary. The x-axis is the number of epochs, and the y-axis is the increased error rate after adversarial attack. We see that instance wise adversary is much stronger compared to universal adversary even though universal adversary achieves competitive performance on clean and ood tasks.

**Attack Strength**   Given that the Universal Pyramid Adversarial training achieves similar performance gain compared to the Pyramid Adversarial training, one may expect that their attack strength is similar. However, we found that the sample-wise adversary is significantly stronger than the universal adversary even though the universal adversary uses a larger radius (8/255 vs. 6/255). In Figure 3, the multi-step adversary consistently achieves a much higher adversarial error rate throughout the training process. The observation shows that we don't necessarily need to make the attack very strong to benefit from adversarial training.

**Qualitative Differences in Perturbation Pattern**   In Figure 4, we visualize the universal and sample-wise adversarial patterns used during training. We find the perturbations to have qualitatively different patterns despite their similar effectiveness in improving clean accuracy.

For the perturbation at the coarser scales, universal perturbation has a more diverse color than sample-wise perturbation. The diversity may be because universal perturbations need to transfer between images, and large brightness changes may be effective in removing some information from the samples. On the other hand, the sample-level perturbations may exploit the color cues to move an image to the adversarial class by consistently changing the color of the image.

For the perturbation at the pixel level, sample-wise perturbation is much more salient compared to universal perturbation. We can see that the sample-wise perturbations have some resemblance to objects. Even though the universal perturbations have some salient patterns, they are less obvious than the pattern from the sample-wise perturbations.

**Loss Landscape**   To understand how Pyramid Adversarial training and Universal Pyramid Adversarial training improve the performance of a model, we visualize the loss landscape of models trained with both approaches to see whether it achieves the performance by implicitly inducing a flatter minimum Li et al. (2017). Surprisingly, we found this not to be the case. Sample-wise Pyramid Adversarial training produces sharper minima compared to regular training and yet has better performance (see Figure 2). On the other hand, Universal Pyramid Adversarial training does not noticeably change the sharpness of the minimum and yet produces the greatest performance improvement. This finding suggests that both adversarial pyramid approaches rely on different underlying mechanisms to improve the model's performance compared to an optimizer, such as SAM Foret et al. (2020) that explicitly searches for flatter minima.

## 5   Conclusion

In this paper, we propose Universal Pyramid Adversarial training to improve the clean performance and out-of-distribution robustness of ViT. It obtains similar accuracy gain as sample-wise Pyramid Adversarial

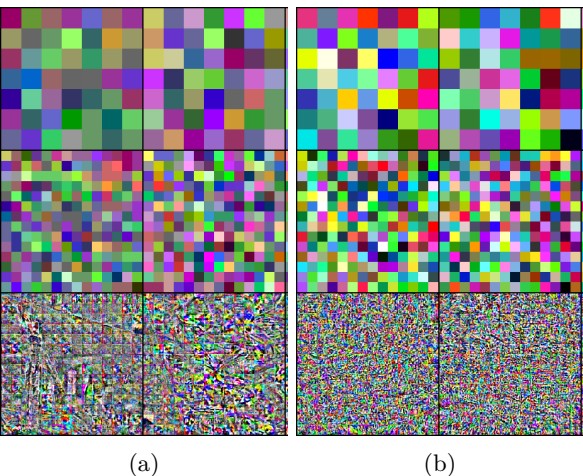

Figure 4: Adversarial patterns generated by sample-wise pyramid adversarial training in 4a and universal pyramid adversarial training in 4b. We plotted each level of the pyramid separately, so that we can visually inspect the differences between each level. We found that (1) universal pyramid seems to rely on more diverse perturbation values at the coarser scale and (2) sample-wise pyramid adversarial seems to rely more on the pixel level perturbations according to the more salient pixel level perturbations.

training while being up to 70% faster than the original approach. To the best of our knowledge, we are also the first to identify universal adversarial training as a possible technique for improving the model's clean accuracy. We hope that the proposed method will help make the adversarial technique more accessible to practitioners and future researchers.

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

# A    Further Details for Table 2

| Method | Radius | Clean ↑ | C ↓ | A ↑ | Rendition ↑ | Sketch ↑ | Stylized ↑ |
|---|---|---|---|---|---|---|---|
| Universal Pyramid Adversarial Training | 12/255 | 80.04% | 40.37% | 28.21% | 37.05% | 17.42% | 19.53% |
| Universal Pyramid Adversarial Training | 8/255 | 80.28% | 41.02% | 29.08% | 35.45% | 16.89% | 17.74% |
| Regular Training | | 79.84% | 42.18% | 27.52% | 33.65% | 15.08% | 17.06% |
| Pyramid Adversarial Training 1 steps | 6/255 | 79.46% | 40.39% | 26.48% | 35.34% | 16.56% | 19.00% |
| Pyramid Adversarial Training 2 steps | 6/255 | 79.87% | 39.76% | 28.49% | 36.17% | 16.95% | 19.95% |
| Pyramid Adversarial Training 3 steps | 6/255 | 80.19% | 38.91% | 30.09% | 37.75% | 17.66% | 20.96% |
| Pyramid Adversarial Training 4 steps | 6/255 | 80.04% | 39.22% | 30.33% | 38.40% | 18.02% | 21.02% |
| Pyramid Adversarial Training 5 steps | 6/255 | 80.10% | 38.84% | 30.04% | 38.63% | 17.80% | 21.21% |

Table 6: Here we provide further details for the performance of each of the adversarial training methods on different corruption datasets.

# B    Visualization of Attention Operations

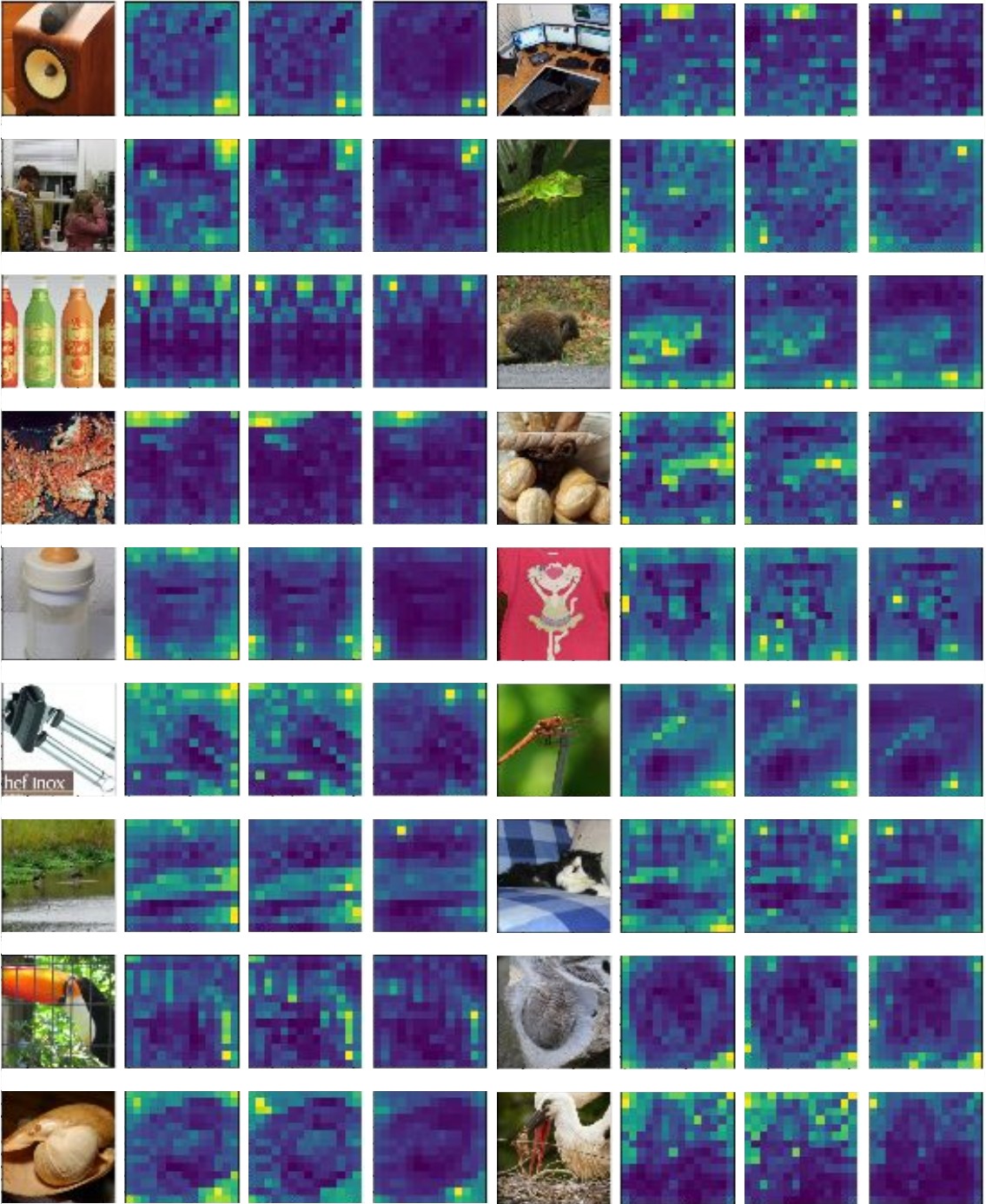

Figure 5: The first column displays the image, followed by attention visualizations for the baseline model, UPAT model, and PAT model in the next three columns. Despite exhibiting similar attention patterns to the baseline model, the UPAT model consistently outperforms it. In contrast, the PAT model demonstrates sparse attention, indicating that UPAT and PAT models improve performance through different mechanisms.

## C  Radius Ablation with Respect to Imagenet V2

| Method | Radius | Top 1 Accuracy (Imagenet v2) | Gain |
|---|---|---|---|
| Baseline | - | 79.81% | - |
| Uni. Pyramid Adv. | 2/255 | 79.93% | 0.12% |
| | 4/255 | 79.91% | 0.10% |
| | 6/255 | 79.94% | 0.13% |
| | 8/255 | 80.24% | **0.43%** |
| | 10/255 | 79.91% | 0.10% |
| | 12/255 | 80.20% | 0.39% |

Table 7: During training, we lacked a separate validation set, raising concerns that our model's hyperparameters may overfit to the test accuracy. To mitigate this, we employed additional evaluation using ImageNet-V2, an independent dataset not used for hyperparameter selection in our experiments. Ultimately, we observed similar evaluation results on ImageNet-V2 and the test set, alleviating concerns of hyperparameter overfitting.

