# OpenReview forum: "Universal Pyramid Adversarial Training for Improved ViT Performance"
_TMLR — Rejected by TMLR_

### Review · Reviewer_gpD1 · 2023-04-23

**Summary Of Contributions:**

In this study, the authors introduce general-purpose pyramidal adversarial training, an effective alternative to pyramidal adversarial training that reduces computational cost by 70% while maintaining most of its clean performance and distribution-shift robustness. Advantage. The experiments focus on the Vision Transformer (ViT) architecture, as it is a general and scalable approach applicable to several domains. The authors make three main contributions: proposing a more efficient training method, evaluating its effectiveness on out-of-distribution datasets, and the "first" identification of universal adversarial training as feasible for improving cleaning performance and robustness technology.

**Audience:**

Yes

**Claims And Evidence:**

Yes

**Requested Changes:**

- For Figure 1, I am unsure if it is good to use the same noise pattern (four noise images) for the Adversarial Pyramid step 2, training step, and Universal Adversarial Pyramid. Also, the noise pattern in Universal Adversarial Pyramid did not show it has been updated.
- it's better to include time/iteration comparisons (Table 3) in Table 2.
- For Table 3, do different models converge using the same training epochs?
- Does the ablation study in Table 5 conducted on the test set, and then reporting test results with the "overfitted" hyper-parameters? The ablation study should perform over validation sets.
- Are there other ways to analyze the learned universal adversarial patterns, rather than Figure 4? For example, compute statistics around the images from a certain class to see how their class is being altered after applying (a) universal noise and (b) sample-specific noise.
- Based on the current experiments, I am unsure how the introduction of pyramid scheme affects performance.
- Table 6 contains both absolute values and relative values, which should be unified.
- I am pretty unsure if this work is the first to "identify universal adversarial training as a viable technique for improving clean performance and robustness".


**Strengths And Weaknesses:**

Overall, I believe this paper is a following up work on universal perturbation and pyramid adversarial training. The combination of the two techniques is natural, and this paper conducted some analysis of the adversarial robustness and out-of-the-distribution robustness. Also, the paper is easy to follow. However, I am unsure if the manuscript is friendly to readers who are not very familiar with adversarial attack/robustness. For example, in Section 3.4, people may get confused about what is radius schedule and how it combines with pyramid adversarial noises.

This paper conducted extensive experiments on standard benchmarks, including ImageNet-1K, ImageNet-C, ImageNet-A, ImageNet Rendition, ImageNet Sketch, and Stylized ImageNet. The results show the Universal Pyramid Adversarial training is generally better than Pyramid Adversarial with 1, 2 steps, while keeping a faster training speed. Also, some ablation studies and analyses are included to examine the proposed framework.

Beside, I mentioned a lot of points in "Requested Changes".

---

> ### Author Response · Authors · 2023-05-10
> **Response**
>
> We appreciate your thoughtful reviews, and we have responded to your concerns below.
>
> **“However, I am unsure if the manuscript is friendly to readers who are not very familiar with adversarial attack/robustness. For example, in Section 3.4, people may get confused about what is radius schedule and how it combines with pyramid adversarial noises.”**
>
> Thank you for pointing this out. We have now updated Section 3.4 to clarify the definition of radius to make it more clear to readers who may not be familiar with adversarial attack/robustness.
>
> **“For Figure 1, I am unsure if it is good to use the same noise pattern (four noise images) for the Adversarial Pyramid step 2, training step, and Universal Adversarial Pyramid. Also, the noise pattern in Universal Adversarial Pyramid did not show it has been updated”**
>
> We have adjusted Figure 1 to clearly demonstrate that the noise patterns are not only different, but also that they are updated throughout the iterations.
>
> **“it's better to include time/iteration comparisons (Table 3) in Table 2.”**
>
> We have incorporated time comparison into both Table 1 and Table 2 given the feedback.
>
> **“For Table 3, do different models converge using the same training epochs?”**
>
> Yes, all of the models were trained with 300 epochs. We have made this more explicit in our updated manuscript in our experimental set up.
>
> **“Does the ablation study in Table 5 conducted on the test set, and then reporting test results with the "overfitted" hyper-parameters? The ablation study should perform over validation sets.”**
>
> We only modified the radius hyperparameter from Table 4, while keeping the rest consistent with Hermann's approach. This reduces the likelihood of overfitting hyperparameters. However, we acknowledged the concern and conducted an additional evaluation using ImageNet-V2, which yielded similar performance results. Although the relative order of evaluated results differs, the baseline model still performs the worst, and the model with a radius of 8/255 achieves the best performance, leading to conclusions consistent with the original findings. We have incorporated these additional evaluations in Appendix C of the updated paper.
>
> | Radius | Test Set | Imagenet V2 |
> |--------|----------|-------------|
> | 0      | 79.84%   | 79.81%      |
> | 2/255  | 79.89%   | 79.93%      |
> | 4/255  | 80.23%   | 79.91%      |
> | 6/255  | 80.15%   | 79.94%      |
> | 8/255  | 80.28%   | 80.24%      |
> | 10/255 | 80.13%   | 79.91%      |
> | 12/255 | 80.04%   | 80.20%      |
>
> **“Are there other ways to analyze the learned universal adversarial patterns, rather than Figure 4?”**
>
> Even though we have not figured out additional ways to analyze the universal adversarial pattern, we have added additional analysis in Appendix B of the attention pattern between the regularly trained model, UPAT model, and PAT model. We found that the attention pattern to be much sparser for PAT compared to UPAT. This suggests that similar benefits can be achieved with UPAT without explicitly sparsifying the attention operation.
>
> **”Based on the current experiments, I am unsure how the introduction of pyramid scheme affects performance.”**
>
> In the update Table 3, you can now see that the introduction of the pyramid scheme is necessary for the performance gain that we observed. Without the pyramid scheme, the model performance decreased as opposed to increase.
>
> **“Table 6 contains both absolute values and relative values, which should be unified.”**
>
> Given the feedback, we have updated the table to have a single column with the absolute and relative values.
>
> **“I am pretty unsure if this work is the first to "identify universal adversarial training as a viable technique for improving clean performance and robustness".”**
>
> It is important to recognize that the positive impact of adversarial training on ViT models has only emerged in research published approximately a year ago. To the best of our knowledge, no existing work has demonstrated the benefits of universal adversarial training specifically for clean and out-of-distribution (OOD) performance. The unique advantage of our approach may be attributed to the incorporation of the recently proposed pyramid structure. As evidenced by our ablation studies, applying plain universal adversarial training without this pyramid structure may actually deteriorate, rather than improve, the final model's performance.

---

### Review · Reviewer_Lbos · 2023-04-27

**Summary Of Contributions:**

This paper addresses the challenge of reducing the cost of a previous training method that aimed to improve clean accuracy and out-of-distribution robustness using adversarial training. It is the first work to enhance ViT's performance on ImageNet1K with adversarial training at a comparably low cost. The proposed method uses universal adversarial perturbations to achieve efficient adversarial training, which can inspire future studies.


**Audience:**

Yes

**Broader Impact Concerns:**

There are no ethical concerns with this work.

**Claims And Evidence:**

Yes

**Requested Changes:**

Besides the weakness stated  above, I recommend the authors to consider the following changes:

- Table 3 can be integrated with Table 1 so it would be clearer about how UPAT is more  efficient and more accurate than the baselines.

- Table 4 shows the gain of UPAT compared to UAT. But the point of this Table is to show that UAT is worse than the standard training baseline. Therefore, I think an explicit comparison with the standard baseline would be helpful.

- Only training errors are presented in Figure 3 but I think the testing errors can further help the readers to better understand the strength of the attacks.

- The paper did not mention the total epochs (and thus the total training time), where I think it is an important hyperparameter to report.

**Strengths And Weaknesses:**

Strengths:

- The paper's objective and motivation are clear.
- The proposed method is simple and somewhat innovative.
- Experimental results demonstrate that the proposed method is effective in reducing training costs and improving, or not harming, performance.
- The paper provides further empirical analysis.

Weaknesses:

- Little intuition and understanding are presented on why Universal Pyramid Adversarial Training improves in- and out-of-distribution performace. All analysis is rather empirical and it would be more valuable if the authors provide more explanations of the phenomena.

- I find it hard to agree with the arguments at the end of the fourth paragraph in Introduction (“In practice, clean accuracy and out-of-distribution robustness are often more critical than adversarial robustness.”). I believe both out-of-distribution robustness and adversarial robustness can be important in some certain situations.

- It is unclear why two sets of experiments are conducted in Table 1 with different data augmentation levels, and why the radius schedule performs worse in the strong data augmentation setting.

- All experiments use ViT-S/16 as the backbone, and it would be more convincing if other base models, such as ViT-B/16 or MLP-Mixer, were experimented with, as in Herrmann et al. (2022).

---

> ### Author Response · Authors · 2023-05-10
> **Response**
>
> We appreciate your thoughtful reviews, and we have responded to your concerns below.
>
> **“Little intuition and understanding are presented on why Universal Pyramid Adversarial Training improves in- and out-of-distribution performance.”**
>
> The improvement in in- and out-of-distribution performance through Universal Pyramid Adversarial Training is difficult to pinpoint precisely; however, we can provide some simple explanations. Generally, the influence of data augmentation on a model's performance is still a topic of debate, as different augmentations can help or hinder its effectiveness. In the case of adversarial training, it is believed that it may strengthen the model by preventing reliance on shortcuts for classification.
>
> By learning from adversarially altered examples, the model becomes more resilient to input data changes and disturbances. This enhanced robustness may lead to improved generalization and performance on out-of-distribution samples. Nevertheless, we admit that our understanding of the exact reason for this benefit remains limited: for example, we found plain universal adversarial training to *decrease* performance whereas universal pyramid adversarial training increases performance. The discrepancies still cannot be fully explained, and we see this as an intriguing area for future investigation.
>
> **“Figure 3 but I think the testing errors can further help the readers to better understand the strength of the attacks.”**
>
> We have revised Figure 3 to include an evaluation of attack strength for both the training and validation sets. This analysis revealed a consistent trend across these datasets. Specifically, we found that instance-wise attacks were substantially more powerful than universal attacks, even though their performance on clean conditions and out-of-distribution (OOD) data remained comparable. This observation highlights the fact that an attack's strength does not always serve as an accurate predictor of a model's ultimate performance.
>
> **“It would be more convincing if other base models, such as ViT-B/16 or MLP-Mixer, were experimented with”**
>
> Unfortunately, due to computational limitations, we were not able to run these experiments. However, we would like to point out that our baseline ViT-S performs (79.8%) comparably to the baseline of ViT-B (79.92%) in Herrmann et al. despite using only a *quarter* of the parameters. This makes our results more credible since we have to increase performance with a smaller capacity model that is already heavily tuned.
>
> **“both out-of-distribution robustness and adversarial robustness can be important in some certain situations.”**
>
> We concur that both adversarial robustness and out-of-distribution robustness can be of great importance in various scenarios. In light of your feedback, we have adjusted the manuscript to present a more balanced perspective on the significance of these robustness aspects.
>
> **“It is unclear why two sets of experiments are conducted in Table 1 with different data augmentation levels, and why the radius schedule performs worse in the strong data augmentation setting.”**
>
> We conducted two sets of experiments with different data augmentation levels in Table 1 to examine the potential benefits of Universal Pyramid Adversarial Training both in conjunction with and independent of strong data augmentation. Strong data augmentation typically provides significant performance improvements for ViT models, and we aimed to investigate whether our proposed method offers additional advantages. Our results indicate that although Universal Pyramid Adversarial Training has some overlap with strong data augmentation, it still provides further benefits on top of it.
>
> Regarding the radius schedule, our initial hypothesis was that it could reduce distribution shift towards the end of training, thereby enhancing performance. However, our observations revealed inconsistent effects, ultimately disproving this hypothesis. Despite these inconsistencies, we identified instances where the radius schedule was indeed beneficial, which prompted us to include the results in Table 1.
>
> **“Table 3 can be integrated with Table 1 so it would be clearer about how UPAT is more  efficient and more accurate than the baselines.”**
>
> We have updated Table 1 and 2 to include training time so that it is more clear to the reader that our approach is more efficient.
>
> **“Table 4 shows the gain of UPAT compared to UAT. … I think an explicit comparison with the standard baseline would be helpful.”**
>
> We have updated Table 4 (now Table 3) to include a more explicit comparison with the baseline.
>
> **“The paper did not mention the total epochs (and thus the total training time), where I think it is an important hyperparameter to report.”**
>
> We have updated 4.1 experimental set-up to include the total number of epochs. We use the same number of epochs across all experiments that is consistent with Hermann et al.

---

### Review · Reviewer_9tSM · 2023-05-01

**Summary Of Contributions:**

In this paper, the authors mainly focus on the issue of Pyramid Adversarial Training (PAT) which is 7 times more expensive than standard training due to sample-wise patterns. The authors propose a Universal Pyramid Adversarial training technique by learning pyramid adversarial patterns shared across the whole dataset. The experiments on several datasets demonstrate that the method is able to decrease the computational cost of PAT and retain the majority of its benefit on clean performance and distribution-shift robustness.

**Audience:**

Yes

**Claims And Evidence:**

Yes

**Requested Changes:**

1. The statement "clean accuracy and out-of-distribution robustness are often more critical than adversarial robustness" needs citations or other supports. In the contributions, "the pyramid structure is critical for the performance gain" was verified in Pyramid Adversarial training (Herrmann et al., 2022).

2. In Eqn. (2), the notion $\lambda$ is used without defined. It is a trade-off parameter of two losses. In Eqn. (3), how to tune the multiplicative constant $m_s$ in the experiments.

3. In Line 11 of Algorithm 1, is the gradient of the loss w.r.t. $x$ or $\delta_s$? Are the objective (2) and (4) equivalent? Could you show the difference?

4. In Table 2, compared with Pyramid Adversarial Training, the proposed Universal Pyramid Adversarial Training is not very significant.

5. In Figure 2, the comparisons of these three examples are very similar, and the flatness of the loss landscape cannot demonstrate whether the performance is good or not.

6. Can the proposed Universal Pyramid Adversarial Training improve the performance of attention? Could you show some visualizations of the attention for different models?

**Strengths And Weaknesses:**

Strengths:
1. The authors make the first attempt to exploit a single pyramid adversarial pattern shared across the whole dataset to improve the performance.
2. The proposed method is 70% more efficient than the multi-step approach.

Weaknesses:
1. Some technical details are not clear. Please refer to the detailed comments below.

---

> ### Author Response · Authors · 2023-05-10
> **Response**
>
> We thank the reviewer for your thoughtful reviews, and we have responded to your concerns below.
>
> **“The statement "clean accuracy and out-of-distribution robustness are often more critical than adversarial robustness" needs citations or other supports.”**
>
> We made the statement due to the anecdotal observation in the industrial setting. However, we understand the statement may be too strong, and we have reworded the statement to acknowledge the different scenarios where adversarial robustness may be useful.
>
> **“In the contributions, "the pyramid structure is critical for the performance gain" was verified in Pyramid Adversarial training (Herrmann et al., 2022).”**
>
> In our study, we highlight the significance of the pyramid structure when it comes to universal adversarial training, distinguishing it from instance-wise adversarial training. We've clarified this distinction in our revised manuscript, particularly in the third point of our contributions, citing Herrmann et al. as a reference. It's noteworthy that universal adversarial training behaves differently compared to adversarial training. For instance, basic universal adversarial training doesn't enhance, but rather undermines, the performance of Vision Transformer (ViT) as compared to adversarial training. This distinct behavior underscores the need to affirm the advantages of the pyramid structure in this new context.
>
> **“In Eqn. (2), the notion  is used without definition. It is a trade-off parameter of two losses.“**
>
> Thank you for pointing this out. We have updated the paragraph after Eqn. (2) to include the definition of $\lambda$
>
> **“In Eqn. (3),  how to tune the multiplicative constant  in the experiments.”**
>
> We did not tune the multiplicative constant in the paper. Rather, we used the exact same hyperparameter as (Herrmann et al.).
>
> **“In Line 11 of Algorithm 1, is the gradient of the loss w.r.t. $\delta$ or $x$”**
>
> The gradient is w.r.t. $\delta_s$ as opposed to $x$, since we are updating the adversarial pattern in line 11, which has a pyramid structure, unlike $x$, which is in the pixel space.
>
> **“Are the objective (2) and (4) equivalent? Could you show the difference?”**
>
> The two objectives are different since (4) optimizes for a single pattern shared across the whole dataset whereas the (2) optimizes a single pattern for each example. In fact, (2) is always a strict upper bound of (4) due to Jensen’s inequality. We have clarified this in the paragraph after equation (4).
>
> **“In Table 2, compared with Pyramid Adversarial Training, the proposed Universal Pyramid Adversarial Training is not very significant.”**
>
> Our paper focuses on the efficiency and effectiveness trade-off. While our proposed approach does not outperform 5-step instance-wise approach, it is quite strong given the same computational constraints.
>
> **“In Figure 2, the comparisons of these three examples are very similar, and the flatness of the loss landscape cannot demonstrate whether the performance is good or not.”**
>
> In Figure 2, we explored the assumption that adversarial training enhances performance by smoothing the loss landscape, a seemingly logical and intuitive hypothesis. However, our experimental results disproved this assumption, as shown by the negative outcomes illustrated in Figure 2. Despite the unexpected results, we see them as valuable contributions for future investigations thus the inclusion of the figure.
>
> **“Can the proposed Universal Pyramid Adversarial Training improve the performance of attention? Could you show some visualizations of the attention for different models?”**
>
> We have now visualized the attention of the baseline model, UPAT model, and PAT model in the updated Appendix B. Despite exhibiting similar attention patterns to the baseline model, the UPAT model consistently outperforms it. In contrast, the PAT model demonstrates sparse attention, indicating that UPAT and PAT models improve performance potentially through different mechanisms.

---

### Decision · Action_Editors · 2023-06-15

**Recommendation:** Reject

**Comment:**

The paper can be viewed as an extension of Pyramid Adversarial training Herrmann et al. (2022). However, the experimental results presented in the paper may not be comprehensive enough to justify the effectiveness of the proposed algorithm. In the work of Pyramid Adversarial training Herrmann et al. (2022), various networks were explored, such as ViT-B/16, ViTTi/16, ResNet, MLP-Mixer, and Discrete ViT. While we acknowledge that the authors' baseline ViT-S performs comparably with the baseline ViT-B/16 in Herrmann et al. (2022), it does not demonstrate that the advantages of the new algorithm can consistently perform as well as expected across different network baselines. Additionally, reviewers questioned why the proposed algorithm can improve in- and out-of-distribution performance. Although the authors provided some explanation, there is a lack of concrete or objective evidence to support their statement. The authors are recommended to undergo a major revision.





**Audience:**

The paper addresses the importance of clean accuracy and out-of-distribution robustness. Though the proposed algorithm itself could be a bit incremental by combining universal adversarial training Shafahi et al. (2020) and Pyramid Adversarial training Herrmann et al. (2022), reviewers thought the resulting algorithm could be effective and believe it can bring benefits to the research community.

**Claims And Evidence:**

Based on the review comments provided, it seems that there is a mixed response from the reviewers. While some concerns have been addressed and the strengths of the paper have been recognized, there are still some remaining concerns, e..g, the limited analysis of the learned universal adversarial patterns, the limited architecture studies, etc.  As the authors claimed that the paper is to improve clean accuracy and out-of-distribution robustness, the reviewer suggests alternative approaches like augmentation and transfer learning instead of adversarial training.



**Resubmission Of Major Revision:**

The authors may consider submitting a major revision at a later time.